# Systematic analysis of cilia characteristics and Hedgehog signaling in five immortal cell lines

**Arianna Ericka Gómez**[1,2], **Angela K. Christman**[1], **Julie Craft Van De Weghe**[1], **Malaney Finn**[1], **Dan Doherty**[1]*

**1** Department of Pediatrics, University of Washington, Seattle, Washington, United States of America,
**2** Molecular Medicine and Mechanisms of Disease PhD Program, Department of Laboratory Medicine and Pathology, University of Washington, Seattle, Washington, United States of America

* ddoher@uw.edu

## Abstract

Dysfunction of the primary cilium, a microtubule-based signaling organelle, leads to genetic conditions called ciliopathies. Hedgehog (Hh) signaling is mediated by the primary cilium in vertebrates and is therefore implicated in ciliopathies; however, it is not clear which immortal cell lines are the most appropriate for modeling pathway response in human disease; therefore, we systematically evaluated Hh in five commercially available, immortal mammalian cell lines: ARPE-19, HEK293T, hTERT RPE-1, NIH/3T3, and SH-SY5Y. Under proper conditions, all of the cell lines ciliated adequately for our subsequent experiments, except for SH-SY5Y which were excluded from further analysis. hTERT RPE-1 and NIH/3T3 cells relocalized Hh pathway components Smoothened (SMO) and GPR161 and upregulated Hh target genes in response to pathway stimulation. In contrast, pathway stimulation did not induce target gene expression in ARPE-19 and HEK293T cells, despite SMO and GPR161 relocalization. These data indicate that human hTERT RPE-1 cells and murine NIH/3T3 cells, but not ARPE-19 and HEK293T cells, are suitable for modeling the role of Hh signaling in ciliopathies.

## Background

Primary cilia are microtubule-based organelles that transduce light, mechanical, and chemical signals into cells [1]. Protruding from the surface of most cell types, primary cilia mediate multiple signaling pathways, including Hedgehog (Hh), that play important roles in development and tissue homeostasis. Cilium dysfunction results in conditions called ciliopathies, and while abnormal Hh signaling has been described in models of ciliopathy-associated genes, details of the disease mechanism have not been clearly elucidated [2]. Immortal cell lines are frequently used to study primary cilium biology and the mechanisms underlying ciliopathies [3–6]; however, the strengths and weaknesses of commonly used human cell lines have not been systematically compared. The goal of this study was to identify immortal cell lines suitable for studying Hh signaling in human health and disease.

In vertebrates, the primary cilium modulates the canonical Hh signaling pathway through the coordinated translocation of Hh pathway proteins in and out of the cilium (reviewed in

**Funding:** This work was supported by the National Institutes of Health (https://www.nih.gov/grants-funding). AG is supported by the University of Wahington Molecular Medicine Training Program T32GM095421 and an NIH Diversity Supplement under parent grant U54HD083091. JCV is supported by grant 5K99HD100554. DD is supported by grant R01HD100730 and P50HD103524 UW IDDRC Genetics Core. The funders had no role in study design, data collection and analysis, decision to publish, or preparation of the manuscript.

**Competing interests:** The authors have declared that no competing interests exist.

[7]). In the absence of Hh ligand, ciliary localization of the Patched (PTCH1) receptor indirectly represses ciliary localization of Smoothened (SMO), maintaining the pathway in an inactive state. This repression is reinforced by ciliary GPR161, an orphan constitutively active G-coupled protein receptor and negative regulator of the pathway, which maintains high ciliary cyclic AMP (cAMP) levels which in turn increases protein kinase A (PKA) activity. Increased PKA activity leads to proteolytic cleavage of GLI transcription factors into their repressor forms, maintaining expression of target genes low [8]. During pathway stimulation, Hh ligand binds PTCH, promoting its export from the cilium, which results in SMO entry and GPR161 exit. In concert, GLI2/3, KIF7, and Suppressor of Fused (SUFU) localization to the ciliary tip increases, and is associated with reduced GLI2/3 cleavage [9, 10]. Uncleaved GLI2/3 activator forms translocate to the nucleus and induce transcriptional targets including *GLI1* and *PTCH1* [11, 12]. In this work, we focus on canonical Hh signaling because it has been implicated in the mechanisms underlying ciliopathies across multiple model systems. We compare two steps in the Hh pathway in immortal cell lines: initial ciliary localization of Hh pathway components and downstream Hh target gene expression.

## Materials and methods

### Cell culture

Cell lines were obtained from the American Type Culture Collection (ATCC, www.atcc.org) and grown in cell specific medium. ARPE-19 (ATCC, CRL-2302) and hTERT RPE-1 (ATCC, CRL- 4000) cells were cultured in DMEM/F12 with 10% fetal bovine serum (FBS) and 1% penicillin-streptomycin. HEK293T (ATCC, CRL-3216) cells were cultured in DMEM with 10% FBS and 1% penicillin-streptomycin. NIH/3T3 (ATCC, CRL-1658) cells were cultured in DMEM with 10% bovine calf serum and 1% penicillin-streptomycin. SH-SY5Y (ATCC, CRL-2266) cells were cultured in 1:1 Eagle's Minimum Essential Medium:F12 with 10% FBS and 1% penicillin-streptomycin. 0.05% trypsin was used for all cell dissociation.

### Time course assay

For time course experiments, cells were seeded on pairs of coverslips coated with 0.3mg/mL poly-D-lysine and allowed to grow for 2 days, until they reached 60–80% confluency. We define batches as cells that were grown, treated, and collected at the same time. For ciliation experiments, coverslips were grown to the desired density, then all coverslips were serum starved. Serum starvation included removing the complete growth medium, rinsing with PBS, and adding serum free medium. At the time of serum starvation, the coverslips for time point 0 were fixed. Coverslips for all other time points were fixed after they were serum starved for the desired length of time (8, 16, 24,. . . etc hours). At each time point, we placed the plate holding the coverslips on ice for 10 minutes, then fixed coverslips using 4% paraformaldehyde for 5 minutes followed by permeabilization using cold methanol for 3 minutes and stored in PBS at 4˚C until ready to stain. We only stained one coverslip from the pair, leaving the replicate available as a backup.

For cilium length experiments, we seeded coverslips in pairs and allowed cells to grow for 2–3 days to the desired density. We then serum starved the first pair of coverslips for the longest time point (96- or 72-hours). We repeated this step for each consecutive time point, so that all coverslips from the same cell line and batch were collected at the end of the time course (0-hour serum starvation). At the end of the experiment, we placed the plates on ice for 10 minutes before fixing, permeabilizing, and staining cells as described above.

For HEK293T cells, we incubated the 0.3 mg/mL poly-D-lysine coated coverslips with 0.05% gelatin for 15–20 minutes prior to seeding cells to promote adhesion since this cell line

is semi-adherent. Otherwise, we followed the same seeding and fixation method described above. The HEK293T isolates used in S2 Fig were from different freezes of the same cell line. We excluded areas of multilayer growth so that we could reliably count cilia and measure semi-quantitative immunofluorescence.

Plots were generated using GraphPad Prism version 9.4.1 for MacOS, GraphPad Software, San Diego, California USA, www.graphpad.com.

### Hedgehog pathway protein localization

Cells in the same batch were seeded on pairs of coverslips and allowed to expand until they reached 60–80% confluency. Cells were serum starved for a total of 48 hours. After the initial 24 hours of serum starvation, we replaced the media on half of the coverslips with serum free medium + 1μM Smoothened Agonist (SAG) (Millipore, 566661). We placed the plates holding the coverslips on ice for 10 minutes, then fixed coverslips using 4% paraformaldehyde for 5 minutes followed by permeabilization using cold methanol for 3 minutes and stored in PBS at 4˚C until ready to stain. We only stained one coverslip from the pair, leaving the replicate available as a backup.

Plots were generated using Plots of Data and Super Plots of Data [13, 14].

### Semi-quantitative immunofluorescence

We blocked coverslips in 2% bovine albumin serum (BSA) in phosphate buffered solution (PBS) for 20 minutes at room temperature. Time course assay coverslips were incubated with mouse anti-acetylated α-tubulin (1:2000, Sigma Aldrich, T6793) and rabbit anti-ARL13B (1:800, Proteintech, 17711-1-AP) antibodies diluted in blocking buffer. Hedgehog pathway protein localization coverslips were incubated with either 1) mouse anti-SMO (1:50, Santa Cruz Biotechnology, sc-166685) and rabbit anti-ARL13B (1:800, Proteintech, 17711-1-AP) antibodies or 2) mouse anti-ARL13B (1:200, NeuroMab, 75–287) and rabbit anti-GPR161 (1:200, Proteintech, 13398-1-AP) antibodies diluted in blocking buffer. Coverslips were incubated with primary antibodies for 1 hour at room temperature or at 4˚C overnight, then washed in PBS for 5 minutes, three times. Coverslips were incubated with the following secondary antibodies at 1:400 dilution for 1 hour at room temperature: Goat anti-Rabbit IgG, Alexa Fluor 488 (Thermo Fisher Scientific #A11008) and Donkey anti-Mouse IgG, Alexa Fluor 568 (Thermo Fisher Scientific, #A10037). After incubation, we washed coverslips in PBS for 5 minutes, three times. Coverslips were mounted on slides using a Fluoromount with DAPI (Invitrogen, #00-4959-52), then sealed with nail polish after sitting for at least 1 hour.

We quantified ciliary protein localization using a validated protocol previously established in the laboratory [15]. We imaged coverslips using the same microscopy settings to acquire z-stack images for >20 cells with cilia for each condition and batch using identical microscope settings. We converted z-stack images to sum-projections and randomized these images using the FIJI [16] script, Filename_randomizer (https://imagej.nih.gov/ij/macros/Filename_Randomizer.txt) to minimize bias between cell lines and conditions. Prior to data collection, we checked a subset of images to ensure that the cilium marker (ARL13B) and protein of interest (SMO or GPR161) signals were adequate for measurement. Blinded to the condition, we drew a cilium mask in the cilium marker channel (ARL13B) using summed Z-stack images in ImageJ. We used this mask to measure the signal intensity for the protein of interest and measured the signal intensity of an adjacent region to subtract background signal. To compare ciliary protein content across different batches, we calculated normalized fluorescence intensity using the following formulas:.

Unstimulated cells:

$$\frac{Unstimulated\ cilium\ average\ fluorescence\ intensity\ -\ Unstimulated\ background\ average\ fluorescence\ intensity}{\bar{x}\ Unstimulated\ (Cilium\ average\ fluorescence\ intensity\ -\ Background\ average\ fluorescence\ intensity)}$$

Stimulated cells:

$$\frac{Stimulated\ cilium\ average\ fluorescence\ intensity\ -\ Stimulated\ background\ average\ fluorescence\ intensity}{\bar{x}\ Unstimulated\ (Cilium\ average\ fluorescence\ intensity\ -\ Background\ average\ fluorescence\ intensity)}$$

Finally, we unblinded the data and visually inspected the images to ensure that our qualitative assessment of ciliary signal was consistent with the quantitative data. We also used the same data set to determine proportion of cells with cilia and cilium length in baseline and stimulated cells.

## *GLI1* and *PTCH1* qPCR

Cells were grown in pairs of T-75 cell culture flasks until 60–80% confluent, then starved for a total of 48 hours. After the initial 24 hours, half the flasks had their media replaced with serum free medium + 1μM SAG. We dissociated the cells and extracted RNA using the Aurum Total RNA mini kit (Biorad, 7326820). We measured RNA concentration using a spectrophotometer and only included RNA that had an A260/280 >1.8. cDNA was generated using the BioRad iScript cDNA Synthesis kit. We set up qPCR reactions using the PowerUp SYBR Green Master Mix (Thermo Fisher Scientific, #A25741). For cell lines that required the touchdown qPCR protocol, we followed the protocol outlined in Zhang et al. [17]. Briefly, the touchdown qPCR protocol went as follows: 50C (2 min), 95C (2 min), 4x [95C (20 sec), 65C (10 sec, decrease 3C/cycle, 72C (1 min)], 40x [95C (15 sec), 55C (15 sec), 72C (1 min)], ending with a melt temperature curve. qPCR data acquisition was performed on the Bio-Rad CFX96 Touch Real-Time PCR Detection System.

Human *GLI1* primers

Forward 5'-GATGACCCCACCACCAATCAGTAG-3'

Reverse 5'-AGACAGTCCTTCTGTCCCCACA-3'

Human *PTCH1 primers*

Forward 5'-GAGCACTTCAAGGGGTACGA-3'

Reverse 5'-GGAAAGCACCTTTTGAGTGG-3'

Mouse *Gli1* primers

Forward 5'-CCGACGGAGGTCTCTTTGTC-3'

Reverse 5'-GCGTCTCAGGGAAGGATGAG-3'

Mouse *Ptch1* primers

Forward 5'-GAGCAGATTTCCAAGGGGAAG -3'

Reverse 5'-CCACAACCAAAAACTTGCCG -3'

## Reference gene identification

To ensure robust data acquisition, we identified an ideal reference gene for each cell line. We identified 10 human candidate reference genes and 9 mouse candidate reference genes (Table 1) to evaluate expression stability between baseline and stimulated cDNA. To select a

**Table 1. Candidate qPCR reference gene names and primer sequences.**

| Gene symbol | Gene name | Forward and Reverse Primer |
|---|---|---|
| ACTB | Actin Beta | 5'-GAGCACAGAGCCTCGCCTTT-3' |
| | | 5'-TCATCATCCATGGTGAGCTGG-3' |
| GAPDH | Glyceraldehyde 3-phosphate dehydrogenase | 5'-AGGTGAAGGTCGGAGTCAAC-3' |
| | | 5'-TTCACACCCATGACGAACAT-3' |
| IPO8 | Importin 8 | 5'-TGCAGTCCGGCCTACTGTTC-3' |
| | | 5'-TGTAGGACTGGTTGAGCTCGTTC-3' |
| PUM1 | Pumillio RNA Binding Family Member | 5'-AAGGACAGCAGCAGGTTCTC-3' |
| | | 5'-CCTTGTCCAAATGCAAGGGC-3' |
| RPLP0 | Ribosomal Protein Lateral Stalk Subunit P0 | 5'-CGTCCTCGTGGAAGTGACAT-3' |
| | | 5'-TAGTTGGACTTCCAGGTCGC-3' |
| SDHA | Succinate dehydrogenase complex flavoprotein subunit A | 5'-GCATTTGGCCTTTCTGAGGC-3' |
| | | 5'-TTGATTCCTCCCTGTGCTGC-3' |
| TBP | TATA Binding Protein | 5'-GTGACCCAGCATCACTGTTTC-3' |
| | | 5'-AGAGCATCTCCAGCACACTC-3' |
| UBC | Ubiquitin C | 5'-CCGGGATTTGGGTCGCAG-3' |
| | | 5'-TCACGAAGATCTGCATTGTCAAG-3' |
| YWHAZ | Tryosine 3-Monooygenase.Tryptophan 5-Monooygenase Activation Protein Zeta | 5'-GACACAGAACATCCAGTCATGG-3' |
| | | 5'-TCATATCGCTCAGCCTGCTC-3' |
| 18S [19] | 18S rRNA | 5'-AGAAACGGCTACCACATCCA-3' |
| | | 5'-CACCAGACTTGCCCTCCA-3' |
| Actb | Actin Beta | 5'-TAGGCACCAGGGTGTGATG-3' |
| | | 5'-TCTCCATGTCGTCCCAGTTG-3' |
| Gapdh | Glyceraldehyde 3-phosphate dehydrogenase | 5'-AATGTGTCCGTCGTGGATCT-3' |
| | | 5'-ATACGGCTACAGCAACAGGG-3' |
| po8 | Importin 8 | 5'-ACAAGCTCTGCTGACTGTGC-3' |
| | | 5'-CAGTGTCCTTCGGTGCTCTG-3' |
| Pum1 | Pumillio RNA Binding Family Member | 5'-GAAAGGTAAGGGGGAGCGAG-3' |
| | | 5'-CTCATTCCACCAACACGGGC-3' |
| Rplp0 | Ribosomal Protein Lateral Stalk Subunit P0 | 5'-TCCTCGTTGGAGTGACATCG-3' |
| | | 5'-AGTTGGACTTCCAGGTCGC-3' |
| Sdha | Succinate dehydrogenase complex flavoprotein subunit A | 5'-ACTGTTATTGCTACTGGGGGC-3' |
| | | 5'-CCCTAGTGACCATGGCTGTG-3' |
| Tbp | TATA Binding Protein | 5'-GGTATCTGCTGGCGGTTTGG-3' |
| | | 5'-GAAATAGTGATGCTGGGCACTG-3' |
| Ubc | Ubiquitin C | 5'-CCCACACAAAGCCCCTCAAT-3' |
| | | 5'-AAGATCTGCATCGTCTCTCTCACG-3' |
| Ywhaz | Tryosine 3-Monooygenase.Tryptophan 5-Monooygenase Activation Protein Zeta | 5'-GGTATCTGCTGGCGGTTTGG-3' |
| | | 5'-GAAATAGTGATGCTGGGCACTG-3' |

We included 10 candidate reference genes for human-derived cell lines and 9 candidate gene reference genes for the NIH/3T3 murine-derived cell line.

reference gene, we evaluated the cycle threshold (CT) of three starting primer concentrations, and cDNA dilution curve efficiency [18].

In our initial screen, we were looking for 1) small ΔCT between unstimulated and stimulated within each cell line, indicating unchanged expression level between experimental conditions, and 2) a CT between 20–30 cycles so that expression levels were similar to *GLI1* and *PTCH* expression. We optimized primer concentration by diluting forward and reverse primers sets to three concentrations, 3μM, 5μM, and 7μM. Next, we determined primer efficiency by performing a cDNA dilution series. We followed widely accepted primer efficiency standards, which includes an efficiency of 80–120% and $R^2 \geq 0.99$.

During assay validation, we determined that *GLI1* in ARPE-19, HEK293T, and NIH/3T3 cells amplified at >29 cycles, close to the limit of reliable detection; therefore, we performed a modified qPCR assay in these cell lines, which included a touchdown step prior to the 40-cycle amplification. CT values were counted after the touchdown step. This lowered the detection level by 3 cycles in ARPE-19 and HEK293T cells. In NIH/3T3 cells, we did not see a difference in unstimulated cycle values and only saw a difference of 1 cycle in stimulated cells.

## Statistical analysis

We performed a one-way ANOVA with multiple comparisons to test for significant differences between time 0 and all other time points in cilium length experiments. We performed a Mann-Whitney test to test for significant differences in the proportions of ciliated cells between stimulated and unstimulated conditions. We performed an unpaired two-tailed Student's *t*-test to test for significant differences in ciliary protein levels between unstimulated and stimulated cells with a hypothesized mean difference of 0. α level was set at 0.05. $P < 0.05$ was considered significant. Symbols for significance represent *p*-values from 0.01 to 0.05 (*), 0.001 to 0.01 (**), and <0.001 (***). No symbols are shown when differences were not statistically significant.

## Results

### Cell line selection

We identified candidate models for human Hh signaling by evaluating the following characteristics in commercially available cell lines: human-derived, evidence of ciliation, immortal, required imaging characteristics (adherent, monolayer growth), evidence of Hh pathway response, and genetic tractability. ARPE-19, HEK293T, hTERT RPE-1, NIH/3T3, and SH-SY5Y cells met most criteria (Table 2). We included murine NIH/3T3 cells as a non-human control since these cells are frequently used to model Hh pathway function [6, 20–22].

### Proportion of cells with cilia

While all the cell lines have been reported to ciliate, the time course of ciliation for different cell lines is scattered across studies [23, 24]. Therefore, we aimed to define the optimal serum starvation conditions for Hh signaling experiments by determining the proportion of ciliated cells across time points after serum starvation: 0, 8, 16, 24, 48, 72, and 96 hours. Serum starvation removes growth factors available to cells, causing cells to drop out of the cell cycle and promoting ciliogenesis. For Hh imaging assays, we wanted cells to be in a monolayer, but we were unable to measure the 96-hour time point for HEK293T cells, because they became overgrown when grown for >72 hours under standard conditions.

Baseline ciliation was less than 15% for all cell lines (Fig 1). Serum starvation was associated with >40% higher ciliation rates over baseline in ARPE-19, hTERT RPE-1, and NIH/3T3 cell

**Table 2. Desired cell line characteristics for modeling Hh signaling.**

|  | ARPE-19 | HEK293T | hTERT RPE-1 | SH-SY5Y | NIH/3T3 |
|---|---|---|---|---|---|
| Cell origin | Retinal pigment epithelial cells | Embryonic kidney cells | Retinal pigment epithelial cells | Bone marrow derived Neuroblastoma cells | Murine embryonic fibroblast cells |
| Human-derived | Y | Y | Y | Y | N |
| Ciliated | Y | Y | Y | Y | Y |
| Immortal | Y | Y | Y | Y | Y |
| Adherent | Y | S | Y | Y | Y |
| Monolayer growth | Y | N | Y | N | Y |
| Hh pathway response | Y | Y | Y | Y | Y |
| Gross chromosomal abnormalities | N | Y | N | N | N |
| Genetic tractability | Y | Y | Y | NR | Y |

We searched the literature for evidence of the desired characteristics listed. ATCC often provided information about the following categories: human-derived, immortal, gross chromosomal abnormalities. When considering any known response to Hedgehog signaling, this included assays that measured response differently from the assays discussed in this paper (i.e. GLI3 processing, GLI luciferase assays). Y = Yes, N = No, S = semi-adherent, NR = Not reported.

lines. Ciliation began to peak in these cell lines at 48 hours post-starvation and ranged from 43% in ARPE-19 to 81% in NIH/3T3. In contrast, HEK293T and SH-SY5Y ciliation rates were similar with and without serum starvation.

SH-SY5Y cells ciliated at low proportions through 96 hours of starvation, prompting us to evaluate 120- and 144-hour time points to determine whether ciliation was delayed compared to other cell lines. Even with prolonged starvation, <8% of SH-SY5Y cells had cilia at each time point (S1 Fig).

Proportions of ciliated cells in HEK293T's have been reported as low as 7% [25], while we and others have previously documented proportions of ciliated HEK239T cells >10% [26]. To determine whether the low proportions of cells with cilia was a characteristic of the specific isolate of HEK293T cells, we repeated the time course experiment using three different isolates of HEK293T cells in our collection (S2A Fig). Baseline ciliation rates were higher in the repeated experiments (~20%) and increased to 32–39% by 24 hours. At 48 and 72 hours, the proportion of cells with cilia was lower than at 24 hours. We compared areas of monolayer and multilayer growth, finding that areas of multilayer growth had more cilia (S2B and S2C Fig).

HEK293T cells are widely used in cilium biology, therefore, we decided to continue evaluating this cell line as a candidate for Hh assays. In the Hh experiments described below, we continued with high density monolayer cultures and acquired enough images for robust semi-quantitative immunofluorescence measurements. The experiments below were performed using an isolate that grew to a higher density and ciliated ~15% with serum starvation and pathway stimulation (Fig 1C).

## Cilium length

We measured cilium length across all timepoints (Fig 1B). Cilium length was significantly shorter at 0 hours for ARPE-19, hTERT RPE-1, and NIH/3T3 cells, stabilizing between 8–24 hours, while cilium length was not significantly different between time 0 and other time points in HEK293T and SH-SY5Y cells. Maximum median length for each cell line was 3.0μm for ARPE-19, 2.1μm for HEK293T, 3.2μm for hTERT RPE-1, 2.8μm for NIH/3T3, and 1.4μm for SH-SY5Y cells. Based on both the low ciliation rate and exceptionally short cilia, we did not continue evaluating SH-SY5Y cells for Hh pathway response.

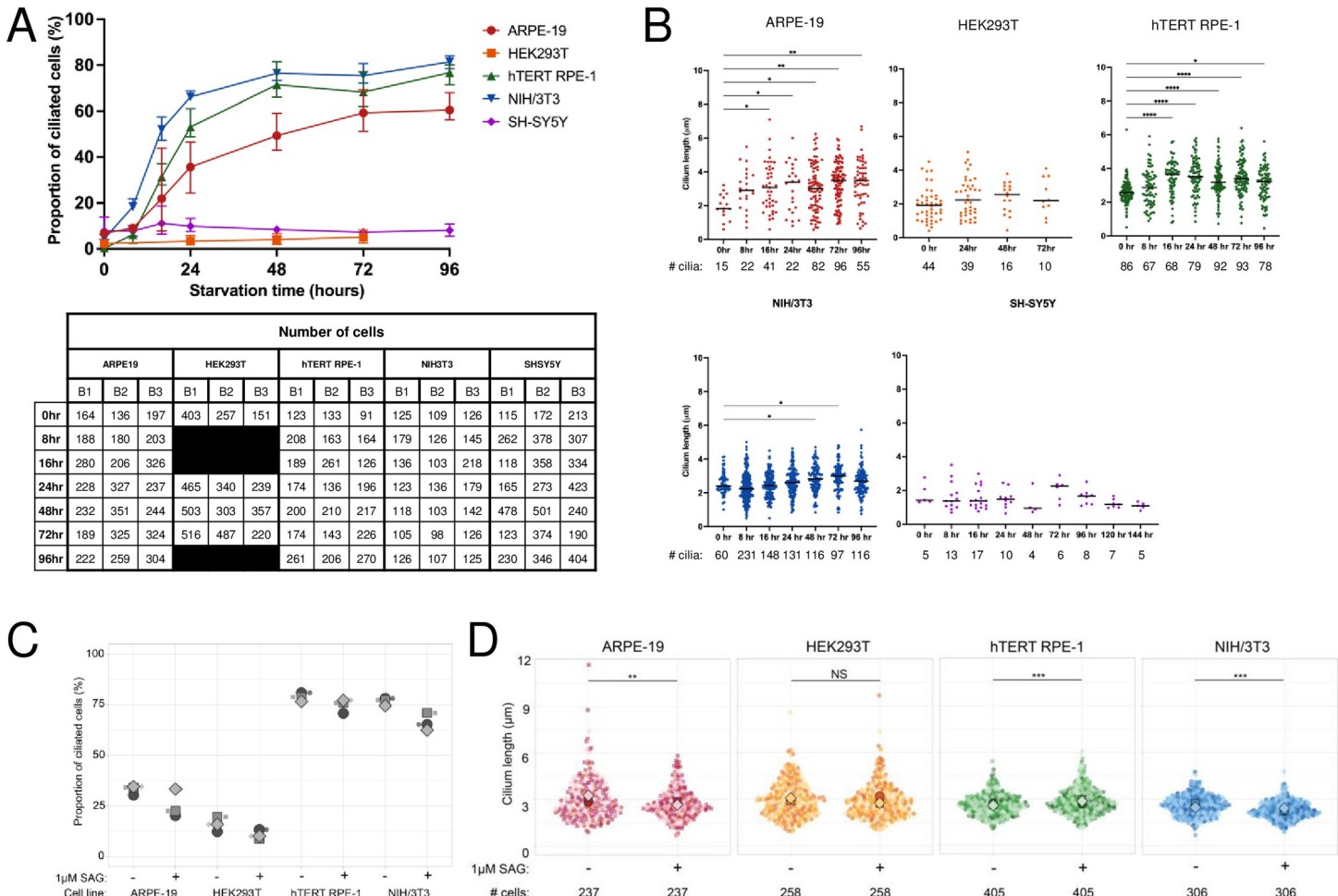

**Fig 1. Ciliation time course and cilia length for immortal cell lines.** A) Proportion of ciliated cells at 0, 8, 16, 24, 48, 72, and 96 hours post serum starvation. We performed three replicates for each time point, with error bars denoting the range for each time point across three batches. The numbers of cells counted at each time point and in each batch are listed in the table below the graph. B1 = batch 1, B2 = batch 2, and B3 = batch 3. B) Cilium length across time points. Black bars in each column represent median length. Significant *p*-values are denoted as follows: 0.01 to 0.05 (*), 0.001 to 0.01 (**), and <0.001 (***). No *'s are shown for differences that are not statistically significant. C) Proportion of ciliated cells without and with stimulation. Each symbol represents the proportion of ciliated cells in one batch. No differences between untreated and treated cells within each cell line were statistically significant using the Mann-Whitney test. D) Cilium length without stimulation and with stimulation. Large symbols = median cilium length within a batch. Small symbols = cilium length of one cilium. One coverslip was imaged for each cell line and time point. The total number of cells counted across three batches are listed below each column of the graphs.

## Hedgehog pathway response

We next measured canonical Hh pathway activity in the cell lines using two Hh pathway assays commonly reported in the literature: 1) upstream localization of Hh pathway effectors SMO and GPR161 into and out of the cilium, respectively [8, 27], and 2) downstream target gene (*GLI1* and *PTCH1)* induction [11].

**Hh pathway protein localization.** We measured the ciliary localization of SMO and GPR161 in unstimulated and Smoothened agonist (SAG) stimulated cells using our established semi-quantitative immunofluorescence (qIF) protocol that determines relative protein localization between different cell lines and conditions [15].

**Proportion of cells with cilia and cilium length in response to Hh stimulation.** Proportion of ciliated cells and cilium length were moderately affected by pathway stimulation (Fig 1C and 1D). Proportion of ciliated cells was lower in stimulated cells compared to

unstimulated cells across cell lines in a majority of batches. Cilium length was significantly decreased in ARPE-19 and NIH/3T3 cells, significantly increased in hTERT RPE-1 cells, and was not significantly different in HEK293T cells.

**SMO localization.** Normalized mean ciliary SMO signal was significantly higher in stimulated cells compared to unstimulated cells across all cell lines, with mean stimulated intensities ranging from: 1.5–2.6-fold in ARPE-19, 1.2–2.7-fold in HEK293T, 2.1–2.9-fold in hTERT RPE-1, and 7.0–10.3-fold in NIH/3T3 cells (Fig 2A). In ARPE-19 and HEK293T cells, SMO ciliary fluorescence intensity was weaker and background signal was higher compared to hTERT RPE-1 and NIH/3T3 cells (S2A Fig).

**GPR161 localization.** We detected cilium-specific GPR161 signal in HEK293T and hTERT RPE-1 cells, a faint cilium-specific signal in ARPE-19 cells, and no cilium-specific signal in NIH/3T3 cells (Fig 2B), so we could not measure GPR161 response in NIH/3T3 cells. Across three batches, normalized mean GPR161 intensity was lower with SAG stimulation:

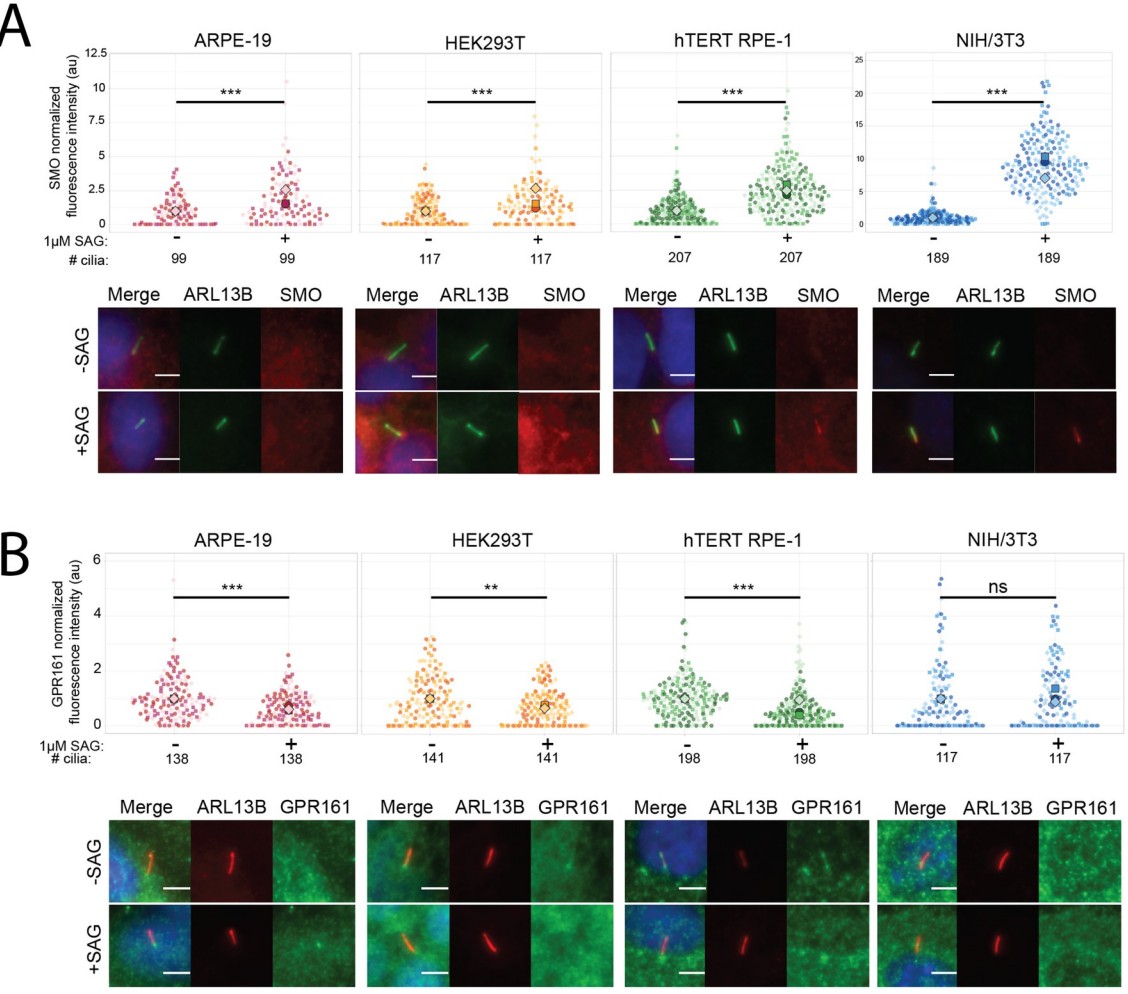

**Fig 2. SMO and GPR161 localization in response to Hh pathway stimulation.** Normalized fluorescence intensity (au) of A) SMO and B) GPR161 with and without pathway stimulation. Large symbol = median normalized fluorescence intensity for individual coverslip data. Small symbol = normalized fluorescence intensity of one cilium. Measurement from three separate batches are indicated by a unique symbol shape and color. Total number of cilia measured listed in the second row below the graph. Representative images are shown below graphs for unstimulated (-SAG) and stimulated (+SAG) cells. Two-tailed Student's *t*-test was performed. *P*-values represented significant differences if they ranged from 0.01 to 0.05 (*), 0.001 to 0.01 (**), and <0.001 (***).

0.57–0.73-fold in ARPE-19, 0.64–0.76-fold in HEK293T, and 0.40–0.95-fold in hTERT RPE-1 (Fig 2B). For one of the three hTERT RPE-1 passages (Batch 3), the GPR161 signal was the same with and without stimulation. We suspect that this was due to an undetermined technical issue since hTERT RPE-1 cells responded when we repeated the experiment on an additional passage (S2C Fig).

**Hh target gene expression.** *GLI1* and *PTCH1* are direct transcriptional outputs of the canonical Hh pathway and are commonly used to measure pathway activity [11]. To measure *GLI1* and *PTCH1* expression, we optimized our quantitative PCR assays using the $2^{-\Delta\Delta CT}$ method [28].

To accurately measure transcriptional responses to pathway stimulation using the $2^{-\Delta\Delta CT}$ quantitative PCR method, we identified suitable reference genes whose expression levels were similar to *GLI1* and *PTCH1* (>20 cycles, <30 cycles) and did not change with pathway stimulation in each cell line. We searched the literature for commonly used reference genes, identifying ten candidates for human cell lines and nine candidates for the mouse cell line (primers listed in Methods). We chose the genes with the most similar expression levels in stimulated and unstimulated cells, with the amplification cycles closest to *GLI1* and *PTCH1*: *PUM1* for ARPE-19 cells, *TBP* and *PMBS* for HEK293T cells, *TBP* for hTERT RPE-1 cells, and *Sdha* for NIH/3T3 cells (S4 Fig). We also used murine specific *Gli1* and *Ptch1* primer sets for NIH/3T3 cells. Although we tested *TBP* and *PMBS* for HEK293T, and used *PMBS* in our experiment, neither was ideal: *TBP* expression was variable in stimulated cells, and *PMBS* expression varied with stimulation in one out of the three batches (S4 Fig).

In hTERT RPE-1 cells, *GLI1* expression was 2.8 to 3.8-fold higher and *PTCH1* expression was 1.7 to 1.9-fold higher with stimulation (Fig 3). In contrast, stimulation was not associated with marked expression differences in ARPE-19 cells (*GLI1* 0.9 to 1.4-fold, *PTCH1* 1.0 to 1.1-fold) or HEK293T cells (*GLI1* 0.5 to 1.7-fold, *PTCH1* 0.3 to 0.7 fold), consistent with the lack of relocalization of SMO and GPR161 with SAG stimulation. Strikingly, *Gli1* expression was 1993 to 3034-fold higher with stimulation in NIH/3T3 cells. Similarly, *Ptch1* expression was also markedly higher (46 to 70-fold).

## Discussion

This systematic evaluation of cilium characteristics and Hh signaling in five commercially available, immortal, mammalian cell lines revealed that hTERT RPE-1 and NIH/3T3 cells are suitable for modeling Hh signaling (Table 3). By using the same assays in different cell types, we determined differences in response to Hh stimulation. Based on our experiments, SH-SY5Y cells were not suitable because of their low ciliation rate and very short cilia. While HEK293T cells originally ciliated at low proportions, we optimized the conditions and HEK293T isolate to yield proportion of ciliated cells >15% throughout our Hh experiments. Stimulated ARPE-19 and HEK293T cells had higher SMO ciliary signal and lower GPR161 ciliary signal, as expected, yet they failed to upregulate the Hh target genes *GLI1* and *PTCH1*. Only hTERT RPE-1 and NIH/3T3 cells ciliated well and showed robust responses to stimulation in both sets of assays. Our experiments also demonstrate that Hh pathway stimulation does not impact ciliation rates and cilium length consistently across batches. We observed a range of responses to stimulation, from minimally responsive (ARPE-19 and HEK293T), to modestly responsive (hTERT RPE-1) to robustly responsive (NIH/3T3) (Table 3). It is not clear whether this represents species-specific or cell line-specific differences, but it does indicate that experiments should be validated across model systems before they can be considered generalizable.

It is also clear from our experiments that each Hh assay requires optimization for each cell line. Prior to assessing Hh response, we had to determine acceptable seeding, growth

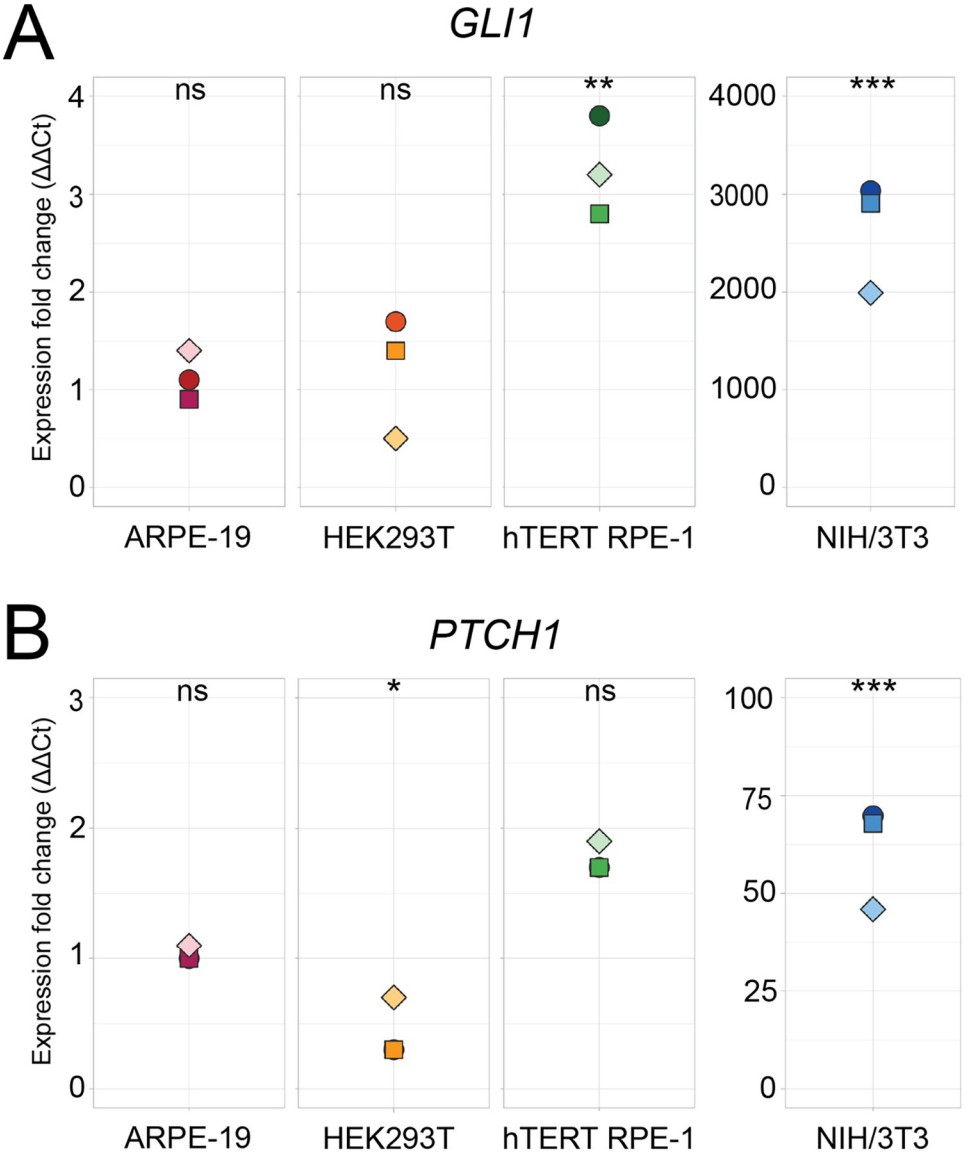

**Fig 3.** *GLI1* **and** *PTCH1* **expression in response to Hh pathway stimulation.** A) *GLI1* and B) *PTCH1* target gene induction. Each data point represents the expression fold change of stimulated cells from the same batch. Expression fold change was normalized to unstimulated values, which were set to 1. Two-tailed Student's *t*-test was performed. *P*-values represented significant differences if they ranged from 0.01 to 0.05 (*), 0.001 to 0.01 (**), and <0.001 (***).

conditions, and fixation. In HEK293T cells, we initially had low proportions of ciliated cells, but changing the seeding density and using a different isolate yielded higher ciliation rates. Even after optimization, we had to balance the upside of higher ciliation rates in confluent cultures with the downside of overlapping cells complicating qIF measurements; therefore, low ciliation rates in HEK293T cells may have contributed to the low Hh target gene induction.

For the qIF experiments, we had to identify antibodies that worked across cell lines and were specific to the protein of interest. The limited response to SAG stimulation in ARPE-19 and HEK293T cells could be due to difference in expression of Hh negative regulators in these cell lines that differ from hTERT RPE-1 and NIH/3T3 cells, for example SUFU expression which complexes with GLI2/3 and stabilizes the activator forms of the proteins [29].

**Table 3. Overview of results.**

|  | Max proportion of cells with cilia | Cilium length (mean) | SMO and GPR161 response to stimulation | GLI1/PTCH1 induction |
|---|---|---|---|---|
| **ARPE-19** | 65% (72 hr) | 3.0μm | SMO: ↑2.50-fold | GLI1: ND |
|  |  |  | GPR161: ↓0.25-fold | PTCH1: ND |
| **HEK293T** | 8% (72 hr)[1] | 2.0μm | SMO: ↑2.50-fold | GLI1: ND |
|  |  |  | GPR161: ↓0.25-fold | PTCH1: ND |
| **hTERT RPE-1** | 80% (72 hr) | 3.1μm | SMO: ↑2.50-fold | GLI1: ↑3-fold |
|  |  |  | GPR161: ↓0.50-fold | PTCH1: ↑2-fold |
| **NIH/3T3** | 97% (48 hr) | 2.9μm | SMO: ↑9.50-fold | GLI1: ↑3000-fold |
|  |  |  | GPR161: ND | PTCH1: ↑60-fold |
| **SH-SY5Y** | 8% (16 hr) | 1.3μm |  |  |

Overview of ciliary characterization and Hh assays. hTERT RPE-1 cells were the most responsive human-derived cell line and NIH/3T3 cells were the most responsive cell line overall. We did not measure SH-SY5Y response to Hedgehog signaling because of low proportion of ciliated cells and short cilia.

[1]In follow-up experiments, proportion of ciliated cells was increased by allowing cells to grow at a higher density.

Alternatively, these cells might respond to native Hh ligand or other agonists, but not SAG. There is prior evidence that hTERT RPE-1 and NIH/3T3 cells relocalize SMO and GPR161 and upregulate pathway target genes in response to Hh stimulation [22, 30, 31]. ARPE-19 cells have been used to determine Hh response to cyclopamine, a down-regulator of the pathway [32]. We could not detect Gpr161 in NIH/3T3 cells by immunofluorescence, despite the 94% amino acid identity between the human and mouse for the antigen used to generate the antibody. The most likely reason may be due to differences between the human and mouse proteins at the key epitopes recognized by the polyclonal antibody, since prior work has detected Gpr161 in NIH/3T3 using other antibodies [8]. Alternatively, Gpr161 expression has been demonstrated in other studies using different stimulation time points [33]. Future work could explore ligand dose-response, and optimal timing for SMO and GPR161 localization in response to Hh stimulation.

qPCR also needs to be tailored to each line. We optimized primer sets, template concentrations, and amplification parameters. Importantly, expression of the commonly used reference gene, *GAPDH*, differed between unstimulated and stimulated HEK293T cells. *GAPDH* expression responds to serum starvation [34, 35], adding to the importance of screening reference genes for each cell line. During *GLI1* primer optimization, we identified high CT values in our cell lines (>30 CT), which introduces variability [36]. To bring the CT values into a better range, we increased template concentration and added a touchdown step in ARPE-19 and HEK293T, lowering the CT values by 3–4 cycles [17]. Unstimulated NIH/3T3 samples also had a CT >30, but the touchdown step did not decrease CT values in this cell line, suggesting the *GLI1* expression is below the level of detection in this cell line and increases substantially with stimulation.

While we confirmed that hTERT RPE-1 and NIH/3T3 cells are appropriate cell lines to model Hh signaling, we have not demonstrated that they recapitulate all aspects of Hh signaling across cell types *in vivo*; therefore, Hh experiments in these cell lines should be validated in additional models, ideally *in vivo*. In addition, we have not fully excluded ARPE-19 and HEK293T cells as potential models, since we did not evaluate all aspects of Hh signaling such as dose-response to ligand, time course, and other important characteristics. If there were compelling reasons to use these cell lines for Hh-related experiments, further characterization might reveal intact Hh responses at different timepoints or different agonist concentrations.

hTERT RPE-1 and NIH/3T3 cells respond consistently to Hh pathway stimulation across two standard assays and are genetically tractable. These cell lines provide models originating

from two organisms and could be used individually or in parallel to study Hh response. hTERT RPE-1 cells are a good model system for exploring these mechanistic details at the cellular level.

## Supporting information

**S1 Fig. Extended SH-SY5Y ciliation time course.** Proportion of ciliated cells for SH-SY5Y cells from 0–144 hours. One coverslip was imaged for each timepoint, with number of cells counted at each time point listed below each column in the graph.
(TIF)

**S2 Fig. HEK293T ciliation data.** A) Proportion of ciliated HEK293T cells at 0, 24, 48, and 72 hours post serum starvation from additional thaws. B) Representative images of monolayer areas of HEK293T cell growth from Fig 1, Batch 3, which had the highest ciliation rates (top row) and repeat experiments with additional thaws of HEK293T cells. Cilia were stained with anti-ARL13B antibody. C) Representative images displaying ciliation in areas with multi-layer growth from additional thaws of HEK293T cells. Cilia were stained with anti-ARL13B antibody. Scale bars are 10μm.
(TIF)

**S3 Fig. GPR161 and SMO localization.** Representative images from each batch showing variation in A) SMO and B) GPR161 localization. C) Normalized fluorescence intensity (arbitrary units, au) of GPR161 with and without stimulation in an additional passage of hTERT RPE-1 cells. Black bars = median normalized fluorescence intensity. Small symbol = normalized fluorescence intensity of one cilium. Numbers of cilia measured are listed below each column in the graph. Representative cilia are shown next to the graphs for unstimulated (-SAG) and stimulated (+SAG) cells. Scale bars are 5 μm. T-test of equal variances was performed. p-values: * 0.01 to 0.05, ** 0.001 to 0.01, and *** <0.001.
(TIF)

**S4 Fig. Variable *GLI1*, *PTCH1*, and reference gene expression in response to Hh pathway stimulation.** *GLI1*, *PTCH1*, and reference gene CT (cycle threshold) for ARPE-19, HEK293T, hTERT RPE-1, and NIH/3T3 cell lines with and without stimulation. For cell lines with high CT values (>29), we used a touchdown qPCR method. For each condition (with or without stimulation), we collected mRNA from three batches for each cell line. Normalized *GLI1* and *PTCH1* expression for each qPCR condition is on the far right. Each symbol in the left panels represents the average CT from three technical replicates for each batch. Each symbol in the right panels represents the expression fold change for each batch.
(TIF)

**S1 Data. Experimental data sets.**
(XLSX)

## Author Contributions

**Conceptualization:** Arianna Ericka Gómez, Angela K. Christman, Julie Craft Van De Weghe, Dan Doherty.

**Formal analysis:** Arianna Ericka Gómez.

**Funding acquisition:** Dan Doherty.

**Investigation:** Arianna Ericka Gómez, Angela K. Christman, Malaney Finn.

**Methodology:** Arianna Ericka Gómez, Julie Craft Van De Weghe, Dan Doherty.

**Project administration:** Arianna Ericka Gómez, Dan Doherty.

**Supervision:** Arianna Ericka Gómez, Dan Doherty.

**Validation:** Arianna Ericka Gómez, Malaney Finn.

**Visualization:** Arianna Ericka Gómez.

**Writing – original draft:** Arianna Ericka Gómez, Dan Doherty.

**Writing – review & editing:** Arianna Ericka Gómez, Angela K. Christman, Julie Craft Van De Weghe, Malaney Finn, Dan Doherty.

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
