## [Decision Letter · Decision Letter 0]

27 Apr 2022

PONE-D-22-08157Systematic analysis of cilia characteristics and Hedgehog signaling in five immortal cell linesPLOS ONE

Dear Dr. Gomez,

Thank you for submitting your manuscript to PLOS ONE. After careful consideration, we feel that it has merit but does not fully meet PLOS ONE’s publication criteria as it currently stands. Therefore, we invite you to submit a revised version of the manuscript that addresses the points raised during the review process.

We look forward to receiving your revised manuscript.

Kind regards,

Hemant Khanna

Academic Editor

PLOS ONE

Journal Requirements:

Reviewers' comments:

Reviewer's Responses to Questions

**Comments to the Author**

1. Is the manuscript technically sound, and do the data support the conclusions?

Reviewer #1: Yes

Reviewer #2: Partly

2. Has the statistical analysis been performed appropriately and rigorously? 

Reviewer #1: Yes

Reviewer #2: Yes

3. Have the authors made all data underlying the findings in their manuscript fully available?

Reviewer #1: Yes

Reviewer #2: Yes

4. Is the manuscript presented in an intelligible fashion and written in standard English?

Reviewer #1: Yes

Reviewer #2: Yes

5. Review Comments to the Author

Reviewer #1: Gomez et al have performed a side-by-side analysis of ciliogenesis and Hedgehog (Hh) signaling in five immortal cell lines of interest to the cilia/Hh community. I believe the data are of good quality and will be a valuable addition to the scientific literature. I only want to mention a few points for the authors to consider before publication:

• Figures 1B-C: No statistical analysis is shown. Are the changes significant relative to 0 hours (Fig.1B)? or to no SAG (Fig.1C)?

• Line 137: the equation is somewhat confusing as it is. The terms (Un)Stimulated and x-Unstimulated: are they multiplying the parentheses? If they are just the conditions under which the parentheses are calculated, then it may be better to put them as subindices at the end of the parentheses.

• Line 189: when performing touchdown, could you please clarify if the Ct values are counted starting from the first touchdown cycle, or after the touchdown?

• Line 247: methods should ideally include some information regarding how those HEK293T isolates were obtained.

• Lines 282-3 & 301-2: it would be more meaningful to state fold increases/reductions for each cell line, rather than intensities in arbitrary units.

• Discussion (lines 376-381): it may be worth noting that Mukhopadhyay et al 2013 Cell did report finding Gpr161 in NIH3T3 cilia (see page 211), albeit using a different antibody. Together with several reports of Gpr161 in MEF cilia, this suggests that unstimulated cilia of mouse fibroblasts do contain Gpr161, so the failure to detect it here with the proteintech antibody is likely due to its human specificity, rather than to timing of SAG stimulation, as later proposed (Gpr161 should be there without stimulation, as the authors know).

• Discussion: Akhshi & Trimble 2021 JCB show that SAG can induce ciliogenesis in RPE1 under certain conditions. How does this relate to the observed effects of SAG on ciliation and cilia length in RPE1? This may be worth discussing. Also, since all cell lines in this study were plated on poly-L-lysine, the potential effects of adhesion conditions on ciliation and Hh signaling may also deserve mention.

• Line 50-on: PTCH > PTCH1 ?

• Line 55: lead > leads.

• Line 68: American Type Cell Collection > American Type Culture Collection.

• Line 165: human > mouse.

• Line 221: monolayer layer > monolayer.

• Line 264: localization > translocation ?

• Line 272: for the title, perhaps just say “ciliogenesis” or “ciliation” instead of “proportion of cells with cilia” ?

• Line 275: cell > cells

• Line 343: localizaed > relocalized or translocated.

• Line 355: perhaps remove “we summarized the”?

• Line 356: response > responsive.

Reviewer #2: The manuscript submitted by Gómez and colleagues is a described as a systematic evaluation of cilia and Hedgehog signaling in five commercially available, immortal mammalian cell lines. In the last decade, the cilia field has exploded as this small extracellular signaling organelle is being studied across a broad range of topics from signaling, structural biology, molecular motors, membrane traffic, developmental biology, sensory neuroscience, etc. The authors offer a side-by-side comparison of ciliation/ciliary length/Hedgehog signaling capacity for commonly used cell culture lines. This analysis could be an important reference for this fast-growing field; however, it appears that much of the data presented was only collected from a single coverslip. While these immortalized cell lines are not likely changing much across passages, the seeding density of each coverslip (Methods states was “60-80% confluency”) can significantly alter % ciliation (see Stuck MW et al., Current Biology 2021 for reference). Therefore, to consider this manuscript for publication ALL of the % ciliation experiments should be conducted where a minimum of 3 separate coverslips are analyzed (this was done for HEK293cells in SupFig 2, but not presented for any of the other cell lines. This would be a more rigorous analysis and allow readers to appreciate how cell density might factor into rate of ciliation. I also want to note that due to the variability in sensitivity, specificity, and reproducibility; immunofluorescence is best considered semi-quantitative and not quantitative. Finally, the results section of the manuscript was difficult to read as there were too many headings/subheading and Figure legends were placed in the middle of the text. The revised version should be better presented to reviewers and future readers of PLOS ONE.

Minor Comment:

The first 2 citation of ciliary reviews (1) and (2) are both over 10 years old and should be updated to more recent reviews.

6. PLOS authors have the option to publish the peer review history of their article (what does this mean?). If published, this will include your full peer review and any attached files.

Reviewer #1: No

Reviewer #2: No

---

## [Author Response · Author response to Decision Letter 0]

27 Oct 2022

Response to reviewers

Thank you to the reviewers for their comments on our manuscript “Systematic analysis of cilia characteristics and Hedgehog signaling in five immortal cell lines.” Please find our response to the reviewers below.

Reviewer #1: Gomez et al have performed a side-by-side analysis of ciliogenesis and Hedgehog (Hh) signaling in five immortal cell lines of interest to the cilia/Hh community. I believe the data are of good quality and will be a valuable addition to the scientific literature. I only want to mention a few points for the authors to consider before publication:

Figures 1B-C: No statistical analysis is shown. Are the changes significant relative to 0 hours (Fig.1B)? or to no SAG (Fig.1C)?

• We performed statistical analysis for the data in 1B and 1C, and changed the text in the “Statistical Analysis” section as follows: 

“A one-way ANOVA with multiple comparisons was performed for comparison between time 0 and all other time points in cilium length experiments. A Mann-Whitney test was performed for comparison of stimulated and unstimulated proportions of ciliated cells within each cell line. An unpaired two-tailed Student’s t-test was performed for comparison between unstimulated and stimulated cells with a hypothesized mean difference of 0. α level was set at 0.05. P < 0.05 was significant. Symbols for significance represent p-values from 0.01 to 0.05 (*), 0.001 to 0.01 (**), and <0.001 (***). No symbols are shown if there is no statistical significance.”

• We have also adjusted the figure legend for Figure 1B and 1C to read as follows: 

“B) Cilium length across time points. Black bars in each column represent median length. p-values from 0.01 to 0.05 (*), 0.001 to 0.01 (**), and <0.001 (***). No symbols are shown if there is no statistical significance. C) Proportion of ciliated cells without and with stimulation. Each symbol represents the proportion of ciliated cells in one batch. No statistical significance was found using the Mann-Whitney test between untreated and treated cells within each cell line.”

Line 137: the equation is somewhat confusing as it is. The terms (Un)Stimulated and x-Unstimulated: are they multiplying the parentheses? If they are just the conditions under which the parentheses are calculated, then it may be better to put them as subindices at the end of the parentheses. 

• We separated the formulas used in our semi-quantitative experiments. We now present one formula for unstimulated cells and one for stimulated cells [markup lines 155-158].

Line 189: when performing touchdown, could you please clarify if the Ct values are counted starting from the first touchdown cycle, or after the touchdown?

• Thank you for this suggestion. We now indicate that CT values were counted starting after the touchdown. 

Line 247: methods should ideally include some information regarding how those HEK293T isolates were obtained.

• We now define the origin of the HEK293T isolates in the methods section [markup lines 106-109]. 

Lines 282-3 & 301-2: it would be more meaningful to state fold increases/reductions for each cell line, rather than intensities in arbitrary units.

• We agree that changing the units used from “au” to “fold” is more meaningful, since the normalization to controls represents fold change. 

Discussion (lines 376-381): it may be worth noting that Mukhopadhyay et al 2013 Cell did report finding Gpr161 in NIH3T3 cilia (see page 211), albeit using a different antibody. Together with several reports of Gpr161 in MEF cilia, this suggests that unstimulated cilia of mouse fibroblasts do contain Gpr161, so the failure to detect it here with the proteintech antibody is likely due to its human specificity, rather than to timing of SAG stimulation, as later proposed (Gpr161 should be there without stimulation, as the authors know). 

• Thank you for this suggestion. We revised the manuscript which now reads: “The most likely reason may be due to differences between the human and mouse proteins at the key epitopes recognized by the polyclonal antibody, since prior work has detected Gpr161 in NIH/3T3 using other antibodies.” and we cite Mukhopadhyay et al 2013. 

Discussion: Akhshi & Trimble 2021 JCB show that SAG can induce ciliogenesis in RPE1 under certain conditions. How does this relate to the observed effects of SAG on ciliation and cilia length in RPE1? This may be worth discussing. Also, since all cell lines in this study were plated on poly-L-lysine, the potential effects of adhesion conditions on ciliation and Hh signaling may also deserve mention.

• Thank you for bringing this work to our attention. In Akhshi and Trimble, 2021, the authors were focused on identifying how non-canonical Hh activation can induce ciliation and predominantly used Shh conditioned media. A subset of experiments did use SAG, but there was no evidence that SAG induced higher ciliation rates in cycling cells. Our experimental conditions differed from those in Akhshi and Trimble, since we performed our experiments under serum starved conditions and showed in Figure 1C that the addition of SAG in serum starved cells did not significantly affect proportions of cells with cilia (we now note that a statistical test was performed in figure legend 1C). 

• We note that coated coverslips could affect ciliation and/or Hh signaling, which could contribute in some part to differences between other reports in the cilia. Within our experiments, it was not a factor that was evaluated. 

We made the following minor revisions, as suggested by the reviewer. 

Line 50: PTCH > PTCH1 ?

Line 55: lead > leads.

Line 68: American Type Cell Collection > American Type Culture Collection.

Line 165: human > mouse.

Line 221: monolayer layer > monolayer.

Line 264: localization > translocation ?

• We measured fixed time points, we can only comment on localization, not translocation. 

Line 272: for the title, perhaps just say “ciliogenesis” or “ciliation” instead of “proportion of cells with cilia” ?

• We agree with the reviewers suggestion. 

Line 275: cell > cells

Line 343: localizaed > relocalized or translocated.

• We adjusted the wording in the manuscript to read “Stimulated ARPE-19 and HEK293T cells had higher SMO ciliary signal and lower GPR161 ciliary signal, as expected, yet failed to upregulate the Hh target genes GLI1 and PTCH1.”

Line 355: perhaps remove “we summarized the”?

Line 356: response > responsive.

Reviewer #2: The manuscript submitted by Gómez and colleagues is a described as a systematic evaluation of cilia and Hedgehog signaling in five commercially available, immortal mammalian cell lines. In the last decade, the cilia field has exploded as this small extracellular signaling organelle is being studied across a broad range of topics from signaling, structural biology, molecular motors, membrane traffic, developmental biology, sensory neuroscience, etc. The authors offer a side-by-side comparison of ciliation/ciliary length/Hedgehog signaling capacity for commonly used cell culture lines. This analysis could be an important reference for this fast-growing field; however, it appears that much of the data presented was only collected from a single coverslip. 

While these immortalized cell lines are not likely changing much across passages, the seeding density of each coverslip (Methods states was “60-80% confluency”) can significantly alter % ciliation (see Stuck MW et al., Current Biology 2021 for reference). Therefore, to consider this manuscript for publication ALL of the % ciliation experiments should be conducted where a minimum of 3 separate coverslips are analyzed (this was done for HEK293cells in SupFig 2, but not presented for any of the other cell lines. This would be a more rigorous analysis and allow readers to appreciate how cell density might factor into rate of ciliation. 

• We had a delay in resubmission partially due to the time take to train personnel to work with the five described cell lines after the departure of the primary author from the lab. We repeated the time course experiment using 3 passages of each cell line, which we agree provides more rigor to the study. For this experiment, we modified the protocol (methods, “Time course assay”) to better control for confluency by serum starving all the coverslips at once and fixing cells in order of ascending time points (8, 16, 24, etc. hours). Rather than being collected at the end of the experiment, coverslips for time 0 were collected when all other coverslips for the experiment were serum starved. This brought ciliation at time 0 below 10% for each cell line which is likely caused by a shorter growth time than in the original experiments. In HEK293T experiments, cells ciliated at <7% across batches and time points in the updated time course assay. While we have not been able to determine the reason for this change, some factors we are considering are the new protocol, differences in confluency/density for the isolate used, and changes in reagent lots (the original experiments were performed in 2020). 

I also want to note that due to the variability in sensitivity, specificity, and reproducibility; immunofluorescence is best considered semi-quantitative and not quantitative. 

• We agree with the reviewers comment and edited our use of “quantitative immunofluorescence” to “semi-quantitative immunofluorescence” in the manuscript.

Finally, the results section of the manuscript was difficult to read as there were too many headings/subheading and Figure legends were placed in the middle of the text. The revised version should be better presented to reviewers and future readers of PLOS ONE.

• We agree the headings and figure legend placements can make the draft difficult to read, but the current version of the manuscript is formatted based on the journal requirements.

---

## [Editor Report · Decision Letter 1]

9 Nov 2022

Systematic analysis of cilia characteristics and Hedgehog signaling in five immortal cell lines

PONE-D-22-08157R1

Dear Dr. Gomez,

We’re pleased to inform you that your manuscript has been judged scientifically suitable for publication and will be formally accepted for publication once it meets all outstanding technical requirements.

Kind regards,

Hemant Khanna

Academic Editor

PLOS ONE
---

## [Editor Report · Acceptance letter]

19 Dec 2022

PONE-D-22-08157R1 

Systematic analysis of cilia characteristics and Hedgehog signaling in five immortal cell lines 

Dear Dr. Gómez:

I'm pleased to inform you that your manuscript has been deemed suitable for publication in PLOS ONE. Congratulations! Your manuscript is now with our production department. 

Kind regards, 

on behalf of

Dr. Hemant Khanna 

Academic Editor

PLOS ONE